# Advances in Biodegradable 3D Printed Scaffolds with Carbon-Based Nanomaterials for Bone Regeneration

**DOI:** 10.3390/ma13225083

**Published:** 2020-11-11

**Authors:** Sara Lopez de Armentia, Juan Carlos del Real, Eva Paz, Nicholas Dunne

**Affiliations:** 1Institute for Research in Technology/Mechanical Engineering Dept., Universidad Pontificia Comillas, Alberto Aguilera 25, 28015 Madrid, Spain; sara.lopez@comillas.edu (S.L.d.A.); delreal@comillas.edu (J.C.d.R.); 2Centre for Medical Engineering Research, School of Mechanical and Manufacturing Engineering, Dublin City University, Stokes Building, Collins Avenue, Dublin 9, Ireland; 3School of Mechanical and Manufacturing Engineering, Dublin City University, Dublin 9, Ireland; 4School of Pharmacy, Queen’s University of Belfast, 97 Lisburn Road, Belfast BT9 7BL, UK; 5Department of Mechanical and Manufacturing Engineering, School of Engineering, Trinity College Dublin, Dublin 2, Ireland; 6Advanced Manufacturing Research Centre (I-Form), School of Mechanical and Manufacturing Engineering, Dublin City University, Glasnevin, Dublin 9, Ireland; 7Advanced Materials and Bioengineering Research Centre (AMBER), Trinity College Dublin, Dublin 2, Ireland; 8Advanced Processing Technology Research Centre, Dublin City University, Dublin 9, Ireland; 9Trinity Centre for Biomedical Engineering, Trinity Biomedical Sciences Institute, Trinity College Dublin, Dublin 2, Ireland

**Keywords:** biodegradable scaffolds, bone tissue engineering, carbon-based nanomaterials, additive manufacturing

## Abstract

Bone possesses an inherent capacity to fix itself. However, when a defect larger than a critical size appears, external solutions must be applied. Traditionally, an autograft has been the most used solution in these situations. However, it presents some issues such as donor-site morbidity. In this context, porous biodegradable scaffolds have emerged as an interesting solution. They act as external support for cell growth and degrade when the defect is repaired. For an adequate performance, these scaffolds must meet specific requirements: biocompatibility, interconnected porosity, mechanical properties and biodegradability. To obtain the required porosity, many methods have conventionally been used (e.g., electrospinning, freeze-drying and salt-leaching). However, from the development of additive manufacturing methods a promising solution for this application has been proposed since such methods allow the complete customisation and control of scaffold geometry and porosity. Furthermore, carbon-based nanomaterials present the potential to impart osteoconductivity and antimicrobial properties and reinforce the matrix from a mechanical perspective. These properties make them ideal for use as nanomaterials to improve the properties and performance of scaffolds for bone tissue engineering. This work explores the potential research opportunities and challenges of 3D printed biodegradable composite-based scaffolds containing carbon-based nanomaterials for bone tissue engineering applications.

## 1. Bone Defect Healing

### 1.1. Natural Process of Bone Healing

Bone is composed of a mineralised organic matrix and cells. The matrix provides the bone with its mechanical properties and is comprised of organic and inorganic phases. In the organic phase, type I collagen is the major component and it is responsible for the tensile properties of the bone. Conversely, the inorganic phase comprises hydroxyapatite, which is responsible for exhibiting the compressive properties and for providing the building blocks for new bone formation. Cells are embedded in the matrix, which includes osteoblasts, osteoclasts, osteoprogenitor cells and mature osteocytes [1].

Bone fractures and segmental bone defects are often caused by traumatic injury, cancer or other diseases (e.g., osteoporosis or arthritis) [2]. When a defect appears, the spontaneous fracture healing process begins through two different mechanisms, depending on the mechanical environment. If the strain across the fracture site is less than 2%, primary or direct healing by internal remodeling occurs, whilst secondary or indirect healing by callus formation takes place when strain is between 2 and 10%. The latter type of healing is the process that most fractures follow and it depends on osteogenesis, osteoinductivity and osteoconductivity [3].

However, sometimes bones defect cannot heal spontaneously. This situation occurs especially in large segmental bone defects when the defect reaches a critical size [4]. The precise critical size depends on several factors (i.e., anatomic location, age of the patient, soft tissue environment); in the literature, it is suggested to include defect lengths greater than 1–2 cm and greater than 50% loss of the circumference of the bone [5]. In these cases, additional surgical interventions that help and allow bone healing are required. The available healing processes depending on the size defect are summarized in Figure 1.

### 1.2. Conventional Surgical Solutions

Among the conventional surgical solutions, it is important to highlight bone grafting, distraction osteogenesis and induced membrane techniques [4].

Autologous [6], allograft [7] and synthetic bone grafting [8] are extensively used for repairing bone defects. More specifically, autograft represents the gold standard for the treatment of critical-sized bone defects since it contains all the characteristics for new bone growth (osteoconductivity, osteogenicity and osteoinductivity). However, significant problems are associated with its use, from donor-site morbidity to a limited amount of donor bone [9,10,11]. Other associated issues, such as failed anastomosis, microvascular thrombosis and infection or progressive deformities, have also been reported [12,13].

Another technique extensively studied since the 1950s is distraction osteogenesis. Ilizarov successfully treated his first patient in 1954, reducing the healing time of tibial non-union [14]. This method is based on the capacity of regeneration under tension that the bone presents naturally. Despite the good results that the Ilizarov technique present [15,16], it also has some disadvantages, such as prolonged treatment times, pin site infection [17,18], pin breakage and the inconvenience and burden of prolonged external fixation, which includes muscle contractures, joint luxation and axial deviation [19,20,21,22].

Finally, the induced membrane technique is a two-stage procedure that combines the use of a temporary poly(methyl methacrylate) (PMMA) cement spacer, followed by bone grafting [23]. In 2000, the first cases using the induced membrane technique were reported [24]. Generally, this technique achieves its purpose; however, some complications have been reported. The most common complications include infection, amputation, malunion, fracture and reoperation and additional bone grafting due to non-union [24,25,26].

### 1.3. Scaffolds for Bone Regeneration

To avoid problems encountered when using conventional methods, the field of bone tissue engineering was developed in the early 1980s [27]. Researchers working in the field of bone tissue engineering have made an important effort for developing 3D porous matrices, known as scaffolds. They are based on guided bone regeneration, the aim of which is bone regeneration and growth along the surface of the scaffold [28].

Scaffolds act as a temporary matrix for the attachment, viability and growth of cells whilst maintaining the structure of the regenerated bone in vivo [29]. The main advantage of bone tissue engineering is the potential elimination of donor scarcity, pathogen transfer and immune rejection [30].

Ideally, to properly promote bone regeneration, scaffolds should meet specific requirements [29,31,32]:(1)The material and its degradative by-products should be biocompatible and not evoke inflammation or toxicity when implanted in vivo.(2)Three-dimensional structures should be manufactured in a reproducible manner.(3)High surface area is needed for cell–polymer interactions, extracellular matrix regeneration and minimal diffusion constraints. It is achieved with a porosity of at least 90% and pore size of at least 100 μm [29]. Furthermore, it should have an interconnected porous structure, with a pore size suitable to allow cell adhesion, growth, vascularisation of the tissue and transportation of nutrients.(4)Scaffolds should be capable of being resorbed once their function of providing a template for regenerating bone has completed. Permanent foreign materials inside the body could lead to a permanent risk of inflammation.(5)The degradation or the resorption rate and the rate of bone formation should be similar. For this reason, the degradation rate of the scaffold should have the potential to be adjustable depending on the cell type.(6)Scaffolds should also demonstrate mechanical properties similar to bone.

It is important to highlight that porosity and mechanical properties have an inverse relation. For this reason, a compromise must be found between these characteristics.

### 1.4. Limitation of Bone Tissue-Engineered Scaffolds

Despite the great research advancements in the design, manufacture and application of bone tissue-engineered scaffolds for bone repair and replacement, there still are some drawbacks and challenges that need to be addressed. The main drawbacks of synthetic scaffolds are: poor biodegradability, potential toxic degradation of by-products, poor osteoconductivity, poor mechanical properties, uncontrolled porosity or complicated reproducibility [33].

## 2. Nanomaterials for Scaffolds

### 2.1. Why Are Nanomaterials a Potential Solution?

The use of nanocomposite biomaterials in bone tissue engineering has emerged to improve the mechanical properties as well as physicochemical properties of the polymeric matrix, such as mechanical strength and Young’s modulus, hydrophilicity or biological response (e.g., cell adhesion, proliferation and differentiation, biocompatibility and antimicrobial effect). Nanocomposites for biomedical applications normally have two phases: a biocompatible matrix and a nanosized bioactive/resorbable filler [34,35,36]. One of the main advantages of nanomaterials is their large surface area, which results in a large volume fraction of interfacial material (even at low loadings).

In general, by controlling the volume fraction, arrangement and morphology of the filler phase within the matrix, it is possible to tailor the physicochemical and mechanical properties and the response to the host tissue [27]. In this section, some of the most interesting data relating to the application of nanocomposites in bone tissue engineering will be reported; the application of carbon-based nanomaterials will especially be considered.

### 2.2. Ceramic Nanomaterials

Ceramic nanomaterials, such as calcium phosphates or calcium silicates, may improve the biological response by releasing calcium and phosphate ions that are essential for bone growth. However, a detrimental effect on the mechanical properties has been reported when the amount of inorganic particles is high [37].

The most commonly used ceramic-based nanomaterial is nanohydroxyapatite (nHA). It demonstrates excellent biocompatibility and low toxicity. Several matrices have been filled with nHA (thermoplastic polyurethane (TPU)/polydimethylsiloxane (PDMS), poly ε-caprolactone (PCL), polylactic acid (PLA), etc.) and, in all cases, it has been observed that by comparing the nanocomposite with the pristine matrix, they show a reduction in hydrophobicity, as well as an increase in cell proliferation, mineralisation and differentiation [38,39,40,41].

Calcium phosphate nanoparticles have proved to improve mechanical resistance and the attachment and proliferation of osteoblasts and can demonstrate an antibacterial effect [42,43].

Other ceramics, such as nanosized aluminium oxide [44], titanium oxide [45] or silica [46], have been shown to augment different properties that make them very interesting as potential nanomaterials for bone tissue engineering applications. 

### 2.3. Metallic Nanomaterials

In general, metallic nanomaterials are interesting in bone tissue engineering due to the antimicrobial and bactericidal activity that some of them demonstrate.

Silver nanoparticles are well-known for their antimicrobial activity against a broad spectrum of infectious agents [47]. Specifically, silver ions present a marked antibacterial effect since they cause a disruption of bacteria cell membranes and inhibit enzymatic activities and DNA replication [48]. In the same way, copper and bronze also present bactericidal nature [49].

Gold nanoparticles have also been used to create a polymer nanocomposite due to their inherent low toxicity and antiseptic and antibacterial activity, which prevent bacterial growth in the surgical wound [50,51].

Other metal ions, such as strontium or copper, have been widely used to dope bioactive glasses, improving their osteogenesis, angiogenesis and antibacterial activity [52,53].

Another interesting metal extensively used in bone tissue engineering is magnesium. It presents high mechanical properties, specific strength and elastic modulus, and has a good biodegradability and biocompatibility. However, its high degradation rate limits its application as a matrix material, but the presence of magnesium as a nanomaterial induces osteogenic differentiation [54,55].

### 2.4. Polymer Nanomaterials

Polymers are not used as nanomaterials in bone tissue engineering, but as a coating for other nanomaterials or to enable modification of the matrix. For instance, poly(acrylic acid) or poly(methacrylic acid) grafted to carbon nanotubes improves the potential for cell differentiation of scaffolds [56,57].

In the case of polymers added to the matrix, there are two different paths: co-polymers, formed by two or more monomeric species, and polymer–polymer blends, which involve a mixture of two polymers [27]. Among the co-polymers used in bone tissue engineering, poly(lactic-co-glycolic acid) (PLGA) [58,59] and poly(lactide-co-caprolactone) [60,61] are the most commonly used. Conversely, in the field of polymer blends, many studies are found on gelatin-polyvinyl pyrrolidone [62], gelatin-poly(lactide acid) [63], cellulose acetate-polycaprolactone [64], polyurethane/poly(lactic acid) [65], poly(lactide acid)/polycaprolactone [66,67], etc. Sometimes, they are incorporated into a ceramic matrix to improve their toughness and processability. Both synthetic [68,69] and natural [70] polymers are used for this purpose.

## 3. Carbon-Based Nanomaterials

Carbon-based nanomaterials have shown a high capability for bone tissue engineering since they present excellent mechanical properties and an intrinsic antibacterial activity [71]. The majority of carbon-based nanomaterials have also been shown to promote cell regeneration and decrease the hydrophobicity of the composite material. The interaction of carbon-based nanomaterials with biological molecules depends on the chemical composition, shape, size, stability, functionalisation, porosity and agglomeration of the nanomaterial. Therefore, it is important to study each carbon-based nanomaterial individually [72]. All their biological advantages will be discussed in detail in the next section.

Due to their geometry, carbon-based nanomaterials present an interesting toughening effect that can follow four different mechanisms (Figure 2): (i) crack bridging—carbon-based nanomaterials delay crack propagation, bridging the two surfaces and provide stress that counteracts the applied stress; (ii) pull-out—carbon-based nanomaterials are pulled out the matrix, slowing down crack propagation by the interfacial friction; (iii) crack deflection—crack cannot continue its path for the presence of carbon-based nanomaterials and it follows a tortuous path with high energy dissipation; (iv) crack tip shielding—the crack does not have enough energy for interface debonding and the crack tip is restricted [73].

Thanks to their biological and mechanical improvements, the addition of carbon-based nanomaterials has the potential to transform relatively inert materials into materials demonstrating appropriate mechanical properties and bioresponsiveness for bone tissue engineering scaffolds.

### 3.1. Types of Carbon-Based Nanomaterials

The different types of carbon-based nanomaterials are shown in Figure 3, and explained in the following subsections.

#### 3.1.1. Carbon Nanotubes

In 1991, carbon nanotubes (CNTs) were first reported by Iijima [74], when he discovered multiwalled carbon nanotubes (MWCNTs). Later, in 1993, he discovered single-walled carbon nanotubes (SWCNTs) [75].

CNTs present a tubular shape; their structure is akin to rolled-up graphene sheets. SWCNTs’ diameters are typically in the range 1–2 nm, they are generally narrower than MWCNTs, and they tend to be curved rather than straight [76]. In the case of MWCNTs, the outer diameter depends on the number of layers and ranges from 2 to 30 nm. Typically, the length is in the micrometre range but can differ from 1 µm to a few centimetres [77].

CNTs present unique structural, electrical, mechanical, electromechanical and chemical properties, and as a consequence their application in regenerative medicine has been widely investigated [78].

For example, the addition of MWCNTs to PLA demonstrated an improvement in osteoblast growth [79] due to their excellent electrical properties, since the electrical stimulation may modify cellular activities such as cell adhesion or cell migration [80,81].

Several studies have demonstrated that MWCNTs can promote stem cell differentiation towards bone cells and enhance bone formation. Furthermore, they act as a mode of reinforcement for mechanical strengthening and biocompatibility [56,82,83,84,85], improving the performance of different biomaterials. The case of SWCNTs is very similar, and it has been proved that SWCNTs can be incorporated into some biodegradable polymers that present problems due to their poor mechanical properties to successfully reinforce them [86], without adversely affecting the biocompatibility of the matrix [87].

#### 3.1.2. Graphene Derivatives

##### Graphene

Since its discovery in 2007 [88], the application of graphene (G) has experienced a significant rise. Graphene is a two-dimensional carbon material, one atom thick. Its atomic structure is a honeycomb lattice of carbon atoms and is the basic building block for graphitic materials. It presents a very high surface area (32.216 m^2^·g^−1^), higher than SWCNTs, excellent thermal conductivity (3000–5000 W·m^−1^·K^−1^), mechanical properties (42 N·m^−1^ of break strength, 130.5 GPa of intrinsic tensile strength with Young’s modulus of 1 TPa) and electrical conductivity (5.9 × 10^5^ S·m^−1^) [89,90,91,92].

Since it presents high electrical conductivity, G is used to manufacture electro-active scaffolds. The addition of G stimulates cell proliferation and decreases the immune response [93]. Graphene has also been shown to improve the mechanical properties of biomaterials without compromising biocompatibility [94].

##### Graphene Oxide

Graphene oxide (GO) is produced by oxidising graphite in acidic medium, increasing the hydrophilicity of the surface and creating functional groups (i.e., hydroxyl, epoxy, carboxyl, carbonyl, phenol, lactone and quinone) [95]. It can modulate their characteristics with the degree of oxidation. Graphene oxide is an electrical insulating material because it has disrupted sp^2^ bonding networks due to the presence of functional groups. Their mechanical properties are excellent, but they do not reach the values of G (28–47 GPa of intrinsic strength, with a Young’s modulus of 380–470 GPa) [96,97].

Oxygen functional groups present in the GO structure can experience strong hydrogen bonding interactions with the polymer matrix, improving the interfacial adhesion between them and, therefore, the mechanical properties of nanocomposites [98].

Recent studies have demonstrated that GO promotes cell regeneration, which makes it an interesting material for its application in regenerative medicine. For example, GO has been used to induce piezoelectric behaviour that promotes cell proliferation by generating electrical stimulation [99].

Many studies have reported the use of reduced graphene oxide (rGO) instead of GO in the preparation of nanocomposites [70,100,101]. The reduction in GO leads to the elimination of most oxygen-containing functional groups, thus the sp^2^ structure and, therefore, electrical conductivity are partially restored; consequently, rGO reaches excellent electrical conductivity and high mobility [96,102,103].

##### Graphene Nanoplatelets

Graphene nanoplatelets (GNPs) show a structure similar to idealised G, but in comparison, their production is much more cost-effective. They consist of small stacks of G nanosheets, such as the structure found on MWCNTs, but with platelet shapes. GNPs may be functionalised with different chemical entities, such as carboxyl graphene nanoplatelets [104] or GO nanoplatelets [105].

In the case of carboxyl GNPs, they offer good reinforcement from a mechanical point of view and, due to their large surface area, surface roughness and high protein adsorption, they favour osteoblast cell attachment, proliferation and differentiation [104].

#### 3.1.3. Fullerenes

In 1985, Kroto et al. discovered fullerene [106], the third carbon allotrope, after graphite and diamond. Fullerene presents a structure consisting of sp^2^ carbons in a high symmetric cage. Carbon forms pentagons and hexagons packed in a spherical shape [107].

Fullerenes have been used to reinforce polymers with poor mechanical properties. However, ultrashort carbon nanotubes present better results in terms of mechanical reinforcement [108].

#### 3.1.4. Carbon Dots

Carbon dots are zero-dimensional nanomaterials that are formed by 2–3-parallel graphene sheets. Carbon dots consist of a carbogenic core with oxygen, hydrogen and carbon on their surface. They present an excellent solubility in water, biocompatibility, optical properties, chemical inertness and neglectable toxicity [109,110].

Some research works suggest that carbon dots do not affect cell viability, proliferation, metabolism and differentiation [111]. They are not extensively used in the fabrication of bone tissue-engineered scaffolds [112]. However, for other biomedical applications, such as drug/gene delivery, bioimaging or photothermal and photodynamic therapy, they have been extensively studied [110,113].

#### 3.1.5. Nanodiamond

Carbon atoms in diamond exhibit sp^3^ hybridisation. Four bonds are directed towards the corners of a regular tetrahedron. Diamond is extremely hard due to the rigidity of the three-dimensional network [76]. Nanodiamonds (NDs) were discovered in 1963–1982 [107].

When added to a polymeric matrix, NDs act as a cell growth support and improve the chemical stability and the biocompatibility of the nanocomposite [114,115,116].

### 3.2. Carbon-Based Nanomaterials for Bone Regeneration

Carbon-based nanomaterials are promising candidates for bone repair and regeneration applications due to the exceptional combination of excellent mechanical, thermal and electrical properties along with their proven antimicrobial and cell regeneration capabilities.

In general, carbon-based nanomaterials demonstrate antimicrobial activity. The cellular membrane integrity, metabolic processes and morphology of the microorganisms are affected by the direct contact with carbon nanostructures [118].

The size and surface area of the carbon-based nanomaterials play an important role in the inactivation of microorganisms. The smaller the carbon-based nanomaterial, the higher the surface area, which results in an improvement in its interaction with bacteria [119]. Among carbon nanostructures, fullerenes, SWCNTs and GO and their derivatives have been reported to be the most efficient antibacterial agents [118].

Conversely, nanocomposites with carbon nanomaterials favour cell adhesion and proliferation on their surface and encourage bone growth. It is suggested that this bone regeneration capability is due to different effects—e.g., the increase in surface roughness or the improvement of the hydrophilicity because of the presence of carbon-based nanomaterials. It was also reported that carbon-based nanomaterials present an increase in the protein absorption capability due to their large surface area and ability to interact with proteins [120].

Besides, most of the carbon-based nanomaterials exhibit high electrical conductivity, which opens an interesting avenue of discovery for bone tissue-engineered scaffolds, since it has been reported that electrical stimulation induces osteogenesis and stimulates bone repair. This electrical stimulation has already been used for bone consolidation, incorporation of grafts and treatment of osteoporosis [121,122,123,124,125,126].

#### 3.2.1. Antimicrobial Activity

Contrary to bulk materials, nanomaterials have the capacity to cross cell membranes easily, leading to the destructive effect on the bacteria cell [127]. There are four main bactericidal mechanisms of antimicrobial nanomaterials and carbon-based nanomaterials could promote them:Reactive oxygen species (ROS) generation [128]: cell death is produced by the damage of DNA induced by ROS. These ROS include superoxide anions, hydroxyl radical and hydrogen peroxide.Physical damage [129]: some nanostructured materials present sharp edges that can damage the bacterial cell wall membranes.Binding [130]: loss of cell membrane integrity and efflux of cytoplasmic materials can be caused by binding materials on the bacterial cell wall.Release of metal ions [131]: inhibition of adenosine triphosphate (ATP) production and DNA replication produced by metal ions released into the media may cause the death of cells.

As a consequence of their different characteristics, each type of carbon-based nanomaterial promotes one or several of these mechanisms, acting in different ways.

The antimicrobial activity of CNTs can be related to the formation of cell–CNT aggregates that damage the cell wall of bacteria and release their DNA content [132].

Compared to MWCNTs, SWCNTs present more potent antimicrobial activity. Due to their smaller nanotube diameter, they could have tight contact and penetrate easily in the cell wall. Besides, their higher surface area allows SWCNTs to interact better with the cell surface [118].

Antimicrobial activity is also affected by the length of the SWCNT. It is reported that the longer the CNT, the higher the antimicrobial activity. It is explained by the interactions between cells and the SWCNT; the shorter SWCNT is more likely to be self-aggregated without involving lots of bacterial cells [133].

Conversely, the main antimicrobial mechanism of graphene-related nanomaterials is membrane stress provoked by the direct contact with sharp nanosheets after the cell deposition on graphene-based materials [134].

Furthermore, GO can damage cell membranes via generation of ROS. Thus, the antibacterial mechanism of GO is affected by both physical destruction and chemical oxidation, which results in a decrease in bacterial resistance [135].

In the case of fullerenes, their antimicrobial activity could follow three different mechanisms: (i) internalisation of the fullerenes into the bacteria, which inhibits the energy metabolism, (ii) the impairing of the respiratory chain inhibits bacterial growth and (iii) the induction of cell membrane disruption [118,136]. The hydrophobic surface of fullerenes promotes the interaction with membrane lipids and the intercalation between them [136].

Alternatively, NDs present antibacterial activity since they can form a covalent bond with molecules on cell walls or adhere to intracellular components. This process inhibits vital enzymes and proteins, leading to a rapid collapse of the bacterial metabolism and finally cell death [137].

Accordingly, the use of carbon-based materials to improve the antimicrobial capacity of biomaterials is very interesting and, therefore, their use in the manufacturing of scaffolds may bring great improvements.

#### 3.2.2. Osteoconductivity

First of all, it is important to establish the difference between osteoinductivity and osteoconductivity. Osteoinduction is the process by which osteogenesis is induced—i.e., primitive, undifferentiated and pluripotent cells are stimulated to develop into preosteoblasts. Conversely, osteoconductivity is the capacity of a material to allow bone growth on its surface or down into pores, channels or pipes [138].

CNTs demonstrate osteoconductivity and they can provide a favourable extracellular matrix for cell adhesion due to their similar dimensions to natural collagen fibres. Their osteoconductivity may also be explained by electrochemical interactions between CNTs and cells and by an increase in hydrophilicity [126,139,140].

When graphene-related nanomaterials, especially G and GO, are added to a polymer matrix, it is found that they increase cell adhesion and proliferation by increasing hydrophilicity [141,142,143]. It is reported [144] that this effect is higher in GO, which increases hydrophilicity and leads to better cell behaviour.

However, to achieve osteoinductivity, it is necessary to add some functionalisation to the GO surface, by using, for example, poly(ethylamine) [141], poly(lactide-co-glycolide acid) [145], collagen [146], phosphates [147], hydroxyapatite [148] and silanes [149].

In general, carbon nanomaterials present osteoconductivity, but they are not osteoinductive. However, a recent study [150] has reported that low oxygen content graphene nanoparticles may favour the adhesion of cells and they can undergo osteogenesis on the surface of these nanocomposites.

The case of fullerenes is completely different; fullerene-based films present a decrease in the adhesion of cells with less spreading, growth, metabolic activity and viability. This is explained because fullerene presents an electron-deficiency structure, which is responsible for its high chemical reactivity. For presenting osteoconductivity, fullerene films need to be aged for one year. In that moment, due to fragmentation, oxidation and polymerisation, fullerene supports cell colonisation well [151,152].

Finally, ND incorporation leads to an increase in cell growth on the polymer. As in the case of graphene-related nanomaterials, some studies demonstrated that the wettability is increased with the addition of NDs due to the oxygen termination of the diamond nanoparticles [153,154].

### 3.3. Limitations and Toxicity

In general, carbon-based nanomaterials present inconclusive results when studying their cytotoxicity. The reason is because their toxicity depends on many factors, such as the shape, size, purity, post-production processing steps, oxidative state, functional groups, dispersion state, synthesis methods, route and dose of administration and exposure times [94,155,156]. This makes it very difficult to obtain general conclusions which could be extrapolated to all cases.

Related to the cytotoxicity of CNTs, once implanted in the body, MWCNTs are thought to be biopersistent [157] and could interfere in different physiological processes. For example, in vivo studies reported that the presence of MWCNT agglomerates led to the attachment of multinucleated cells. On its part, SWCNTs were transported from the site of implantation to the lymph nodes and potentially block potassium channel activities in mammalian cell systems. It was also found that the needle-like shape of CNTs promoted the mobility and the penetration of membranes, uptake by cells and strong interactions with various protein systems. These findings may suggest undesirable effects relating to cytotoxicity [158,159,160].

Another problem that can be found in the application of CNTs is that the metal catalysts used in their fabrication are generally trapped inside the nanotubes, which can lead to an increase in cytotoxicity [161].

However, in contrast to these findings, other studies have reported no cytotoxicity effects when CNTs were incorporated into matrix-based materials for bone tissue-engineered scaffolds [162,163,164,165].

Concerning graphene-related materials, the inconsistencies reported from other studies are similar to the studies using CNTs. When they are used as a substrate or coating, some research studies [166,167,168] found that G presented good biocompatibility and the ability to stimulate cell proliferation. Other studies [160,169] reported some cytotoxicity risks since G presented an important tendency to form an agglomerate. When G sheet agglomeration occurs on the cell membrane, it can contribute to their toxic properties. Besides, at high concentrations and long exposure times, ROS were generated by G.

In the case of nanocomposites, contradicting evidence has also been reported when G and GO are incorporated into a matrix. Luo et al. [144] found that the addition of G had a negative impact on cell adhesion, whilst GO presented a positive effect. Türk et al. [170] also found cytotoxicity when G was introduced into a matrix composed of Bioglass. However, several studies [93,143] reported no cytotoxicity when G was introduced in scaffolds (Figure 4 and Figure 5).

In the case of GO, it was found that its presence may reduce the cell proliferation rate due to an increase in the intracellular ROS level [171]. However, other studies [172,173,174,175,176] found that neither GO nor rGO presented cytotoxic effects at low concentrations.

Fullerenes also present contradicting results. On one hand, it was found that fullerene nanoparticles induced DNA breakage [177] and oxidative damage to cellular membranes due to ROS generation [178,179]. However, other studies [180] demonstrated no cytotoxic effect on fullerenes.

In the case of NDs, many studies concluded that no cytotoxicity had been found [181,182,183]. In an extensive study, no ROS generation was found. However, it was found that NDs could easily access the cell membrane, but they seemed to be non-reactive once inside the cell [184].

All these contradicting results make in vitro and in vivo studies crucial for the development and the complete understanding of nanocomposites with carbon-based materials. However, and despite the contradictions found, in most cases, the biocompatibility of carbon-based nanomaterials was found and cytotoxicity was reduced when carbon-based nanomaterials were used as reinforcements embedded inside a matrix.

The area that needs further investigation relates to biodegradable matrices that release carbon-based nanomaterials during their degradation process. Carbon-based nanomaterials were assumed to be biopersistent; however, oxidative enzymes can catalyse the degradation of some carbon-based nanomaterials (e.g., GO or CNTs) [185]. Furthermore, carbon-based nanomaterials may affect the degradation rate of the matrix and the toxicity of the degradation products. To the best of the authors’ knowledge, there are not many research studies relating to this approach.

Sánchez-González et al. [186] found that the addition of rGO caused a slight acceleration in the hydrolytic degradation process, reducing its mechanical stability faster when compared to the pristine polymer. However, rGO was not released to the medium, but it remains embedded in the polymer matrix.

Murray et al. [187] found that when G was added at high levels of loading (5 wt.%), the enzymatic degradation rate was lower than the pristine polymer than the polymer with low levels of G (0.1 and 1 wt.%). They also found that the mechanism by which the G was bonded to the matrix was also affected by the degradation rate. When studying the toxicity of the degradative by-products, it was found that G reduced the toxicity. A decrease in the enzymatic and hydrolytic degradation rate when GO was added was also found. [188,189,190,191]. In the case of enzymatic degradation, the effect of graphene derivatives was due to the presence of sheets that prevented the diffusion of the enzymes into the scaffold network [192].

Finally, Cabral et al. [70] found that neither G nor rGO had a significant effect on the enzymatic degradation rate.

In the case of MWCNTs, Joddar et al. [193] reported a decrease in degradation rate when added to a biodegradable matrix. It may be concluded that the presence of carbon-based nanomaterials incorporated into a matrix reduces the degradation rate and it could be tailored relative to the level of nanomaterial loading.

Table 1 summarizes all the aspects of CBNs regarding their use for bone tissue regeneration.

With these contradictions in mind, it would be interesting to apply concepts to reduce the uncertainties and risks of human and environmental safety surrounding nanotechnology, which have been proposed as an interesting approach during the design of polymeric nanocomposites for biomedical applications. Safe-by-Design (SbD) concepts foresee the risk identification and reduction in early stages of product development. The main elements of the general SbD approach are: (1) stage-gate innovation approach; (2) three pillars—safe materials and products, safe production and safe use and end-of-life; (3) including SbD action for maximizing safety while maintaining functionality and (4) integration into a Safe Innovation Approach [194].

## 4. Additive Manufacturing of Bone Tissue-Engineered Scaffolds

Both material and manufacturing technology affect the final performance of scaffolds. Regarding the manufacturing technology, additive manufacturing has lately been explored and the first results are very promising.

### 4.1. Importance of Porosity and Geometry

The fabrication of bone tissue-engineered scaffolds is associated with the production of controlled porous and interconnected structures since porosity and interconnectivity are two of the requirements needed for achieving adequate bone repair and regeneration. The porosity is important for cell adhesion, growth, revascularisation and adequate nutrition. Kuboki et al. [195] found that solid particles did not promote the formation of new bone, while for porous particles of the same material, osteogenesis occurred. Regarding interconnectivity, Conrad et al. [196] found that interconnectivity increases permeability and, therefore, improves cell infiltration and bone ingrowth.

The size of pores is critical; if pores are not large enough, cell migration is limited, and they may remain at the edges of the scaffold, thereby forming a cellular capsule. Minimum pore size is approximately 100 μm, due to cell size, migration requirements and transport [197]. Conversely, if the pore size is too large, then the surface area decreases and consequently cell adhesion is limited. The results from the study by Murphy et al. [198] corroborate this fact (Figure 6). In extreme cases, for pores that were relatively large or small in size, cell proliferation was not promoted—the cell number remained constant or even decreased. However, intermediate pore sizes promoted cell proliferation over time. Besides, the number of cells was not proportional to the pore size.

Furthermore, pores within a bone tissue-engineered scaffold must be interconnected since this interconnectivity, as well as the pore structure and overall porosity, is fundamental in determining the osteogenic capability of a bone tissue-engineered scaffold [199,200].

Regarding scaffolds’ geometry, there are studies [201,202] that prove how geometry plays an important role in cells response. The geometry may induce stem cells to differentiate towards specific lineage.

Both pore distribution and geometry of scaffolds are decisive in cell penetration, proliferation and differentiation as well as in the rate of scaffold degradation, which must be in accordance with the maturation and regeneration of new tissue.

### 4.2. Additive Manufacturing

To have the appropriate pore size distribution and interconnectivity, different approaches to fabricate porous scaffolds have been studied: salt-leaching [203,204,205], gas foaming [206,207], electrospinning [208,209,210] and freeze-drying [211,212]. However, with these fabrication methods, it is possible to produce bone tissue-engineered scaffolds without controlling pore size distribution and shape, porosity and interconnectivity.

In general, traditional methods present limitations related to the control of overall pore architecture and interconnectivity and bone tissue-engineered scaffolds produced by these techniques present poor reproducibility and accuracy [213,214].

Additive manufacturing (AM) has emerged as a technology that enables the fabrication of 3D porous scaffolds with a high level of reproducibility and accuracy with minimal human intervention.

One of the main advantages of AM techniques is the possibility to produce customised scaffolds with a reproducible internal morphology and pore architecture control, tailored mechanical and mass transport properties and even produce scaffolds with functionally graded materials [215,216].

According to ISO/ASTM standards [217], AM technologies can be classified into seven different groups: (i) binder jetting, (ii) direct energy deposition, (iii) material extrusion, (iv) material jetting, (v) powder bed fusion, (vi) sheet lamination and (vii) vat photopolymerisation.

Outside this classification, other novel techniques are emerging, as is the case of bioprinting, which combine the use of 3D printing technology with materials that incorporate viable living cells. However, these kind of materials fall outside the scope of this review.

Among all the AM technologies, the focus of this review is on those used to manufacture biodegradable scaffolds (i.e., polymer-based and ceramic-based) that allow the incorporation of nanomaterials, more specifically carbon-based nanomaterials.

Researchers have developed biodegradable polymer scaffolds loaded with carbon-based materials fabricated using different AM techniques: (i) material extrusion (e.g., Fused Deposition Modeling, FDM and Direct Ink Writing, DIW), (ii) powder bed fusion (e.g., Selective Laser Sintering, SLS and Selective Laser Melting, SLM) and (iii) vat photopolymerisation (e.g., Stereolithography, SLA and Digital Light Processing, DLP).

In the case of biodegradable ceramic scaffolds, since they do not melt easily, these scaffolds are fabricated by AM using mainly powder bed fusion (SLS). However, to counteract the natural fragility of ceramics, they are usually blended with polymers. For this reason, sometimes other 3D printing techniques are used, such as DIW [54] or FDM [68].

The advantages and limitations of these technologies are detailed in Table 2. Accuracy defines the minimum pore size that can be obtained with each technology. Methods used in each technology for the dispersion of nanoparticles within the matrix are also indicated in the table; this aspect is especially important because the optimal methods to obtain an adequate dispersion are very different depending on the technology.

#### 4.2.1. Material Extrusion

These technologies are based on the extrusion of the material under pressure. The material is deposited from a nozzle or syringe to fabricate components in a layer-by-layer manner. These technologies require a liquid or a viscous material that is obtained mainly by two methods, melting a thermoplastic material (FDM) or using a viscous ink (DIW) [218].

FDM has shown rapid development in recent years due to its simplicity, speediness and large-scale rate of production. Raw materials in FDM are filaments that are partially melted by a heater and extruded from a nozzle.

In the case of DIW, the material used is colloidal ink which is directly extruded through an orifice or nozzle without heating. These inks can maintain their shapes during solidification or drying.

#### 4.2.2. Powder Bed Fusion

SLS and SLM are categorised as powder bed fusion technologies since they utilise thermal energy to selectively melt powder materials of a powder bed. The raw material is typically in the form of powder-based particles for these AM-based technologies [219]. Complete melting is achieved in SLM, while in SLS heat provokes material fusion at the molecular level instead of complete melting [220].

Thermal energy may be obtained by different sources such as lasers or electron beams. In the case of scaffolds with carbon-based nanomaterials, technologies that obtain energy by laser sources are used.

#### 4.2.3. Vat Photopolimerisation

The SLA technique was developed in the 1980s and was one of the first methods proposed for 3D printing [221]. The SLA and DLP techniques involve solidifying a liquid photocurable polymer by exposure to UV light. The photopolymer is placed in a tank and it is cured layer-by-layer on a support platform with a light source (250–400 nm).

The difference between the two techniques is the light source: in SLA, the polymer is cured using a laser that forms each layer point-by-point; in DLP, all the layers are cured at the same time using a matrix of lasers.

Components printed using these AM techniques require a post-curing process (usually, UV exposure + temperature) to achieve their final properties. Considering biomedical applications, this could be used to sterilise the components to be implanted into the body, which could be an additional advantage [222].

Some researchers [223,224] have taken advantage of these AM techniques by modifying a non-photocurable polymer in order to make it photocurable, allowing the use of biodegradable and biocompatible polymers appropriate for bone tissue-engineered scaffolds.

The addition of nanomaterials to improve the properties of additive manufactured structures has been widely investigated. It is well known that the dispersion of nanomaterials is crucial for the attainment of unique properties for the composite. For example, the presence of agglomerates reduces the mechanical properties of the composite [99,189]. Consequently, the method of incorporating the nanomaterials into the matrix is very important for the final composite’s performance.

In the case of AM technologies, the dispersion of nanomaterials depends on the characteristics of matrix materials. When they are solid (FDM, SLS and SLM), usually a solvent is used to disperse nanomaterials easily by ultrasonication [225,232] or mechanical mixing [226]. In the case of materials that melt easily, if their viscosity is low enough, melt mixing can be used as a solvent-free alternative [227,231]. Finally, in the case of powder matrix and powder nanomaterials, they can be mixed using a rotary tumbler [233].

Conversely, DIW, SLA and DLP use liquid raw materials. Ultrasonication is the most common method to achieve the optimal dispersion of nanomaterials in liquid-based raw materials [236,238,239,240], although other mechanical mixing methods have also been used [229]. Sometimes, when the viscosity is high, a solvent is added, which promotes dispersion of the nanomaterial [237].

### 4.3. Biodegradable Materials for 3D Printed Scaffolds

One of the requirements that the materials must meet for being used as tissue-engineered bone scaffolds is biodegradability. Among biodegradable materials that are used with carbon-based nanomaterials, it is possible to distinguish between polymeric and ceramic materials.

#### 4.3.1. Polymer Matrices

Polymeric materials have been widely used for bone tissue engineering applications. They are classified by their origin: (i) natural and (ii) synthetic polymers. Natural polymers include silk fibroin, collagen, gelatin fibrin, elastin, cellulose, alginate, dextran, starch, chitin/chitosan, glycosaminoglycans and hyaluronic acid, among others. Bone scaffolds fabricated using natural polymers by conventional methods have been extensively reported [241,242]. However, the preferred polymers for AM are the synthetic biodegradable ones, and can be classified as follows [243,244,245,246,247]:

Aliphatic polyesters: include polylactic acid (PLA), polyglycolic acid (PGA), poly ε-caprolactone (PCL), poly (lactic-co-glycolide) (PLG) and poly-(3-hydroxybutyrate-co-3-hydroxybutyrate) (PHBV). They usually undergo degradation through hydrolysis of the ester group situated along their backbone with degradative by-products that are acidic in nature. The degradation rate of these polymers is easily controlled because they can be produced with a tailored structure. Their main drawback is their reduced bioactivity. PGA presents a rapid degradation (2–4 weeks), while PLA degradation takes months to years since it is more hydrophobic than PGA. To obtain intermediate degradation rates, PLGA with varying lactide/glycolide ratios are synthesised. PCL can be degraded by different agents—microorganisms, hydrolytic, enzymatic or intracellular agents. Compared with PLA and PGA, PCL has a slower degradation rate.

Aliphatic polycarbonates: they present low thermal stability and reduced mechanical properties. However, their controlled functional characteristics make them very interesting for bone tissue engineering applications. Among them, poly(trimethylene carbonate) (PTMC) has been used for fabricating bone tissue-engineered scaffolds with carbon-based nanomaterials using AM techniques.

Vinyl polymers: poly(vinyl alcohol) (PVA) is one of the most used vinyl polymers. It is a hydrophilic material. Hydrogels based on PVA show good biomechanical features, and they retain a large amount of water.

Polyurethanes (PUs): used for the fabrication of medical devices, especially long-term implants and biomedical products (e.g., cardiovascular catheters and diaphragms of blood pumps). The main disadvantage of PU is the toxicity of its degradative by-products, which can be reduced by using specific prepolymers.

Non-biodegradable polymers: despite their inability to degrade when implanted in the body, some different synthetic polymers have also been used in the manufacture of bone tissue-engineered scaffolds—polyether ether ketone (PEEK), polyvinylidene difluoride (PVDF) and acrylonitrile butadiene styrene (ABS).

#### 4.3.2. Ceramic Matrices

Among ceramic materials, bioceramics exhibit properties, such as biocompatibility, mechanical compatibility, excellent surface compatibility, antithrombus effects, bactericidal effects and good physical and chemical stability, that make these materials suitable for being used for bone tissue-engineered scaffolds [248]. However, their primary ionic and/or covalent bonds make them relatively brittle. This inherent brittleness, together with low ductility, are major drawbacks for bioceramics and therefore limit their application [249].

Additive manufacturing has been investigated as a viable approach for the fabrication of ceramic-based bone tissue-engineered scaffolds by additive manufacturing, the most common ceramics used include:

Tricalcium phosphate (TCP): has good biocompatibility and does not present any cytotoxic reaction. Bone formation is favoured in contact with TCP due to the release of calcium and phosphate ions. Alpha-TCP (α-TCP) demonstrates a greater degree of solubility and a faster rate of degradation when compared to beta-TCP (β-TCP) [248]. The degradation rate of β-TCP is within the same range as the growth of mature new bone [250].

Hydroxyapatite (HA): its structure and composition are similar to the inorganic component of human bones. HA is highly biocompatible, non-toxic, osteoconductive and it gradually merges with the natural bone. Its main drawbacks are its low mechanical properties and relatively low degradation rate. To improve its mechanical strength, there are two paths: (i) use of reinforcements like other ceramics and (ii) fabrication of HA nanobioceramics [251].

Ca-Si-based ceramics: they present good compression properties and controllable degradation rate. They improve the rate of new bone formation and bone regeneration through a gradual release of Si and Ca ions. However, like many other bioceramics, their brittle nature and low toughness hinder their development in load-bearing applications [249].

Diopside, MgCaSi_2_O_6_ (Di): compared with HA and other bioceramics, offers improved mechanical strength. The Ca, Si and Mg containing ionic products from extracts of Di can stimulate osteoblast proliferation at low concentrations. However, its degradation rate is extremely slow [252].

Bioactive glasses (BGs): they can comprise of Na_2_O, CaO, SiO_2_ and P_2_O_5_. They are considered to be the most promising biomaterials in bone tissue engineering applications as they exhibit osteoinductive properties. Another attractive aspect of BGs is that their degradation rate can be regulated with respect to their chemical composition. Their poor mechanical properties (similar to many bioceramics) is the main drawback [249].

## 5. 3D Printed Scaffolds with Carbon-Based Nanomaterials

Other reviews have been reported the application of carbon-based nanomaterials for tissue-engineered bone scaffolds [71,253,254] fabricated by 3D printing [213,246,255,256,257]. Additional reviews have focused on the use of different materials for scaffold fabrication [258,259]. For this review, the focus is placed on biodegradable scaffolds with carbon-based nanomaterials fabricated using AM. To the best of the authors’ knowledge, a review with this focus has not been reported and there is currently a lack of knowledge regarding the effect of carbon-based nanomaterials during the printing process, how the carbon-based nanomaterials are introduced during the printing process and their effect on the final properties of the 3D printed scaffold. Therefore, the effect of carbon-based nanomaterials on scaffold properties has been studied, as well as the feasibility of using AM techniques in the fabrication of bone tissue-engineered scaffolds.

### 5.1. Biodegradable Polymer Scaffolds

• *Material Extrusion*

Due to its simplicity and speediness, FDM is the most used technology when the matrix is a polymer.

Many studies incorporating GO into different thermoplastic matrices have been found. Both biological and mechanical properties of the matrix were improved on the addition of GO. Melo et al. [260] and Unagolla et al. [261] reported improvement in the antimicrobial properties and the enhancement of cellular response. Melo et al. found an 80% increase in bacterial death after 24 h in contact with a PCL/GO scaffold.

Chen et al. [65] found that an increase of 167% in the compressive modulus was achieved when 5 wt.% GO was incorporated into a PLA–thermoplastic polyurethane (TPU) matrix and a maximum tensile modulus increase of 75.5% on the addition of 0.5 wt.% GO. This wt.% loading was also optimal in terms of cellular growth and proliferation of NIH3T3 mouse embryonic fibroblast cells. Belaid et al. [262] studied both mechanical and biological properties and reported an increase of 30% in Young’s modulus when 0.3 wt.% GO was added to PLA. When the proliferation of MC3T3-E1 cells was studied, they concluded that it was not until Day 7 that higher levels of viability were observed for scaffolds containing 0.2–0.3 wt.% GO loading when compared with pristine PLA.

All these studies introduced the nanomaterial using a solvent which was subsequently evaporated to obtain the filament to feed the printer. Figure 7 shows SEM images of PCL scaffolds with and without GO and it can be observed how GO addition did not affect the printing process as relatively similar geometries were achieved in all the cases.

G has also been used to improve the mechanical and biological properties of tissue-engineered bone scaffolds. Wang et al. [93] conducted in vitro and in vivo studies using G-PCL scaffolds for microcurrent therapy. They had previously found [143] an increase in cell attachment and proliferation due to the high surface area, elastic modulus and stiffness of the G-PCL scaffold. In this case, it was not until Day 14 when G at a loading level of approx. 0.8 wt.% demonstrated a statistically significant improvement in proliferation rate. Furthermore, G presented improved mechanical properties (i.e., compressive modulus and compressive strength) and cell affinity when compared to CNTs [263].

Sayar et al. [264] also used G powder as the reinforcement phase and FDM as the AM technique and following printing the scaffold was crosslinked using UV exposition. When a loading level of 3 wt.% G was incorporated into PTMC, an increase in tensile strength (100%) and electrical conductivity was demonstrated, which has potential for electrical stimulation. It was found that cell density, morphology and viability did not differ when compared to the pristine PTMC.

The mechanical properties of ABS scaffolds reinforced with GNPs at a loading level of 4 wt.% were investigated by Dul et al. [227]. They reported an increase in Young’s modulus; however, the adhesion between the matrix and reinforcement was relatively poor, which led to a decrease in tensile strength and strain at break.

Alam et al. [265] worked with a commercial filament of PLA loaded with carbon nanofibres and GNPs. They found that the presence of nanomaterials produced internal porosity, which resulted in a reduced compressive stiffness of 20%. In contrast, carbon nanomaterials improved hydrophilicity and apatite deposition.

Huang et al. [266] found that an addition of 3 wt.% of MWCNTs into PCL increased the compressive modulus, whilst the addition of 0.25–0.75 wt.% of MWCNTs did not affect this parameter. Nanoindentation properties improved from the addition of 0.75 and 3 wt.% CNTs. These improvements occurred due to an increase in polymer crystallinity due to CNT alignment. It was also found that smaller CNTs tended to agglomerate, thereby improving cell attachment and protein absorption. Mimicking natural bone tissue, Huang et al. [267], incorporated MWCNTs and nHA into PCL and found an increase in mechanical properties, cell proliferation, osteogenic differentiation and scaffold mineralisation.

For all studies, the scaffolds fabricated using FDM with carbon-based nanomaterials produced porous structures that were stable and the addition of a nanomaterial did not hinder the printing process.

Another technology derived from material extrusion is Direct Ink Writing. It was more extensively used for polymeric–ceramic matrices than polymeric. Jakus et al. [268] manufactured inks with high levels of G loading in a PLG-based matrix. The tensile modulus increased by 200% with a level of loading of 20 wt.% G. In vitro and in vivo studies demonstrated a good cellular response from approximately 30,000 cells at Day 1 in PLG to more than 50,000 when G was incorporated (Figure 8).

• *Powder Bed Fusion*

Researchers have also shown a high interest in SLS, especially using GO as the nanomaterial. For the application of this technology, different matrices have been investigated. PVA demonstrated very interesting properties for the application of bone tissue-engineered scaffolds. However, its low mechanical properties may be an obstacle in terms of its development. From this point, Shuai et al. [98] achieved an increase in compressive strength, Young’s modulus and tensile strength by 60, 152 and 69% upon adding GO into the matrix. This improvement was obtained for relatively low levels of loadings (2.5 wt.%), and increases beyond 2.5 wt.% resulted in the formation of agglomerates and ultimately the reduction in mechanical properties. Furthermore, they found an increase in cell adhesion and attachment of the GO to the PVA matrix.

Feng et al. [269] focused their work on reducing the degradation rate of PVA. They blended PVA with PEEK, a non-biodegradable polymer, which successfully reduced the degradation rate. Additionally, Feng et al. [269] reported that 1 wt.% GO addition improved the interfacial bonding between PEEK and PVA, which resulted in an improvement in mechanical properties (i.e., increases of 97 and 150% in compressive strength and compressive modulus). An increase in MG63 cell adhesion and proliferation was also reported when 1 wt.% GO was added to the PVA–PEEK matrix.

Shuai et al. [270] reported the improvement in the mechanical properties of GO-poly(L-lactic acid) (PLLA) scaffolds when fabricated using SLS and on adding Ag nanoparticles [271], which also enhanced antimicrobial activity due to the combination of the capturing effect of GO and killing effect of Ag.

Shuai et al. [99] also found that GO could be used to reinforce piezoelectric polymers for bone tissue engineering applications. Scaffolds containing 0.3 wt.% GO reported an improvement in compressive strength (100%) and tensile strength (25%) and cell adhesion was also enhanced by electrical charge excitation. However, although the polymer was not biodegradable, it could be applied in the field of bone scaffolds.

GO nanoparticles have not only been used during SLS technology, Feng et al. [272] achieved improvements for PHBV in terms of tensile strength and compressive strength (i.e., 94 and 52%) when 2 wt.% of ND particles and 1 wt.% of molybdenum disulfide (MoS_2_) nanosheets were added. MoS_2_ improved the dispersion of NDs due to its steric hindrance effect and vice versa and was found to be better dispersed within the matrix when it was added with NDs by the sandwiched octahedral ND particles.

All the studies regarding the polymer-based bone scaffolds with carbon-based nanomaterials when fabricated using SLS technology adopted a similar dispersion method of the nanoparticles—good GO dispersion was achieved using ultrasonication in water or a solvent that was evaporated to obtain powder for printing (Figure 9).

• *Vat Photopolymerisation*

Finally, vat photopolymerisation was the least used AM technique for the fabrication of polymer-based bone tissue-engineered scaffolds containing carbon-based nanomaterials.

Feng et al. [238] introduced 0.5 wt.% G into a commercial PU resin for SLA. G increased the tensile and flexural properties of the matrix. Feng et al. [240] also fabricated DLP samples comprising the same PU resin and 0.5 wt.% GNPs. In this case, the flexural modulus was improved by 14%, while fracture toughness increased by 28%. In both studies, Feng et al. used a solvent-free method to disperse the carbon-based nanomaterials in the matrix using ultrasonication. Consequently, gyroid scaffolds were fabricated by the two vat photopolymerisation technology (Figure 10). In Table 3 most relevant results of polymer biodegradable scaffolds found in literature are provided.

### 5.2. Biodegradable Ceramic Scaffolds

In relation to ceramic-based bone scaffolds, many researchers have focused their studies on biodegradable scaffolds reinforced with carbon-based nanomaterials fabricated using AM technology.

• *Material Extrusion*

In order to use technologies that are not optimally designed for ceramic feedstock, ceramics have been blended with polymers to get the required flowability. One interesting study reported in the literature was conducted by Lin et al. [68], who took advantage of FDM to obtain bone tissue-engineered scaffolds with 50 wt.% of PCL and 50 wt.% of calcium silicate-graphene (CaSi-G). Fluidity limited the amount of G that could be incorporated and at a 10 wt.% loading a relatively poor level of printability was observed. In vitro and in vivo studies found an increase in osteogenesis and cell proliferation by adding CaSi-G to the matrix (Figure 11). To adequately disperse the CaSi and G, a solvent was used and the CaSi-G was dispersed using ultrasonication followed by evaporation.

DIW has also been investigated as a suitable technique for the fabrication of ceramic-based bone tissue-engineered scaffolds. Two approaches have been investigated—the first is focused on the introduction of the nanomaterials into the ink by mechanical stirring [70,174,273,274] before printing the scaffold (Figure 12A) and the second method involves the addition of the nanomaterials into the printed ceramic structure in the form of a coating (Figure 12B) or by infiltration into the pores (Figure 12C) [54,69].

The preferred carbon-based nanomaterials for use with DIW is GO, and the most employed matrix is TCP with any polymer (either natural or synthetic). Wu et al. [54] investigated the potential of applying a coating prepared with a GO–water suspension (20 mg GO/40 mL water). They found that the incorporation of GO improved the proliferation, alkaline phosphatase activity and osteogenic gene expression of human bone marrow stromal cells (hBMSCs). Boga et al. [174] introduced 0.5 wt.% GO into a TCP/alginic acid (AA) matrix and studied the mechanical properties of the resultant scaffold in dry and wet (i.e., simulated body fluid) conditions. The addition of GO increased the Young’s modulus in dry conditions, but it did not affect the compressive strength in dry conditions or the mechanical properties in wet conditions. Furthermore, GO increased the biomineralisation capacity and the alkaline phosphatase activity.

Conversely, rGO exhibited relatively good responses, better even than GO in terms of compressive properties when tested under dry and wet conditions. Cabral et al. [70] incorporated rGO and GO into TCP-gelatin-chitosan scaffolds. rGO was shown to improve calcium deposition and alkaline phosphatase activity. When rGO infiltrated the pores of the scaffold [69], it showed an increase in cell proliferation and did not affect the ionic dissolution of the TCP.

Shah et al. [273] explored the application of two different inks: the first ink was loaded with HA (75 wt.%) and the second ink was loaded with G (60–70 wt.%) and both inks were used in conjunction with PLGA as the matrix. The final composition was 35:35:30, G:HA:PLGA by weight. They found that the composite demonstrated intermediate tensile properties, whilst under compression, the plastic behaviour was similar to that reported for the G-ink. The cellular response was lower than reported for each ink when used separately, but the mixed ink could be used for transition zones between two distinct tissue types.

Finally, DIW was used by Golçaves et al. [274] with a MWCNT-loaded ink (Figure 13). The level of loading for the ceramic (HA) matrix varied from 40–50 wt.% to a 60–40 wt.% of PCL. In terms of compressive properties, only the inclusion of low loading levels of MWCNTs increased the yield strength. Golçaves et al. [274] reported an optimum loading level of MWCNTs (0.75 wt.%) which increased both the mechanical and electrical properties and allowed the potential to apply an electrical stimulus for bone healing purposes. Furthermore, the incorporation of MWCNTs into PCL increased the levels of cell adhesion and spreading.

• *Powder Bed Fusion*

In contrast to the previously discussed AM techniques, SLS allows the application of ceramics without the requirement of adding a polymer since the raw material is already in the form of a powder. Hence, ceramic-based 3D scaffolds must be fabricated using powder bed fusion.

To date, the majority of studies using powder bed fusion technology for the fabrication of bone tissue-engineered scaffolds have involved G or CNT, and surprisingly no studies have reported the use of GO.

Gao et al. [73] and Shuai et al. [275] studied the incorporation of G into a ceramic composite (i.e., nano 58-S bioactive glass and CaSi) for the fabrication of 3D bone scaffolds using SLS. Both studies found that low levels of G loading significantly improved compressive and fracture properties. However, when the loading level was above 0.5 wt.%, these properties decreased due to the inability to adequately disperse the G within the matrix (Figure 14). Furthermore, Gao et al. [73] also studied cell biocompatibility and their scaffolds exhibited good responses in terms of in vitro cell cultures using osteoblast-like cells (i.e., MG-63).

GNPs were also investigated as nanomaterials by Shuai et al. [276] within a Di matrix. Results showed that due to the good dispersion of the 1 wt.% GNPs within the matrix, the mechanical properties were significantly increased (i.e., 102% in compressive strength and 34% in fracture toughness). MG-63 cells presented good attachment and spreading in vitro.

In the case of CNTs, different studies have reported a toughening effect when CNTs were added to a ceramic-based matrix. However, when the amount of nanomaterial was greater than an optimum level of loading (different in each case), the effect was reduced. Liu et al. [277] reported that 3 wt.% MWCNTs presented the most pronounced effect in terms of mechanical properties for BG-based scaffolds. In the case of Di, following the study by Shuai et al. [278], this level of loading was reduced to 2 wt.% MWCNTs. Finally, Liu-Lan et al. [279] found that the best mechanical properties were obtained when 0.2 wt.% CNTs were incorporated into the matrix. For levels of loading greater than 0.2 wt.%, agglomerates of CNTs formed, which resulted in a decrease in the mechanical properties. In all cases, the scaffolds showed good apatite-formation ability and cytocompatibility.

Finally, Liu et al. [280] studied a synergistic effect of 1 wt.% CNTs combined with 1 wt.% GNPs into a Di-based scaffold. The dispersion of both nanomaterials was better due to the combined presence of both nanomaterials; GNPs improved CNT dispersion due to their space hindrance effect and at the same time the GNP dispersion was improved by the tendency of the CNTs to self-align on the surface of GNPs, thereby constructing a 3D network that inhibited their stacking. The CNT-GDP-Di scaffold demonstrated a higher compressive strength and fracture toughness when compared to the same scaffold structure containing only one the of nanomaterials. Furthermore, evidence of good bioactivity and cytocompatibility were also reported.Table 4 provides the main results of ceramic biodegradable scaffolds obtained by additive manufacturing.

## 6. Potential Translability into Clinics

In spite of the promising results obtained from in vitro and in vivo tests, translability into clinics of scaffolds with CBNs requires special attention since it is mandatory to satisfy strict regulatory requirements of US-FDA and EU-MDR for their potential application.

Translation of bone tissue engineering scaffolds to the clinic finds several regulatory hurdles that must be considered from the design stage [281]. Despite these hurdles, there are already some FDA-approved Tissue Engineered products, such as TissueMend^®^, used to repair the rotator cuff, or Osteomesh^®^ for craniofacial repair [282]. However, the introduction of CBNs as a nanofiller in bone tissue engineering introduces additional risks associated with possible toxicity, immunogenicity, cell culture adaptation/morphogenesis, or contamination which must be addressed to assure safety [283,284].

Depending on the material used to manufacture the scaffold, the regulatory process may change. In this respect, there are several FDA-approved synthetic polymers that have been incorporated as structural components in tissue engineering scaffolds. These include polyethylene glycol (PEG), polyglycolic acid (PGA), polylactic acid (PLLA), polycaprolactone (PCL) and their co-polymers [282]. Besides, some ceramic materials are FDA-approved, such as calcium phosphate cements [285].

In the case of the development of biodegradable scaffolds, it may require significant effort to establish the safety of the material as well as its degradation products, increasing time and cost for preclinical and clinical evaluations. The cost and effort required to translate fundamentally new technologies can be a significant barrier.

A substantial amount of theoretical modeling, in vitro characterisation and in vivo (i.e., in animals) studies are needed prior to beginning the regulatory approval process through the filing of an Investigational New Drug (IND) or Investigational Device Exemption (IDE) applications. One concern raised in the field of tissue-engineered scaffolds is the extent to which these studies are predictive of eventual function and performance in humans [286,287].

For the regulatory approval of a new therapeutic entity, the U.S. FDA presents four different pathways: tissues, biological products, drugs and medical devices. Bone tissue engineering finds unique challenges at this stage since, depending on its manifestation, many approaches may fall into more than one category, even all four of these categories. It implies an important difference in times; bringing a device to market takes an average of 3 to 7 years, compared with an average of 12 years for drugs [288,289].

Besides the regulatory approval, there are other business barriers that include obtaining external funding for product development, obtaining physician acceptance and, in some circumstances, obtaining approval for insurance reimbursement [290].

## 7. Conclusions and Future Perspectives

The present work is an overview on very recent developments in the field of biodegradable scaffolds; the aim is to analyze advantages, challenges and potential that the combination of two emerging and promising technologies can provide in this field—additive manufacturing and the use of CBN. Based on the studies, the fabrication of 3D printed biodegradable scaffolds modified with CBN have been shown to be an encouraging solution with interesting benefits—improved mechanical properties, enhanced biological activity, easy control of porosity and design, among others.

However, although considerable progress has been made, thereby providing a promising clinical platform for the repair and regeneration of bone, some aspects need to be examined in greater depth since the technology is still in its infancy. Some of the major research and technical challenges that the scientist community will need to address are:-Good performance of carbon-based nanomaterials is linked to a good dispersion within the matrix, one of the crucial and critical aspects to achieve during the manufacturing of carbon-based nanomaterial-derived bone tissue-engineered bone scaffolds.-It is important to study in-depth the influence that the level of nanomaterial loading exhibits on mechanical and biological properties since there is a balance to be attained to ensure the optimal properties are achieved for both. The optimal level of loading reported depends greatly on the study and ranges from 0.2 to 18 wt.%. In general, using a level of nanomaterial loading less than 1 wt.% offered the best results in terms of mechanical reinforcement.-AM technologies offer many advantages; however, materials need to have specific characteristics to allow the fabrication method to function both effectively and efficiently. The addition of nanomaterials can affect printability and therefore studying and optimising the addition and dispersion methods are crucial for the development of bone tissue-engineered scaffolds reinforced with carbon-based nanomaterials. Further studies regarding how nanomaterials affect the 3D printing technique and how to mitigate possible adverse effects need investigation.-The surface of the carbon-based nanomaterials is easily functionalized, and this functionalisation could improve the dispersion of the carbon-based nanomaterials and ultimately the mechanical performance. Conversely, carbon-based nanomaterials can be biofunctionalised to be used as biomolecular carriers, thereby increasing their bioactivity. Both approaches are interesting and relevant to the fabrication of bone tissue-engineered scaffolds using AM techniques. However, a better understanding of the regenerative effect and bioresponsiveness of chemically functionalised carbon-based nanomaterials and the mechanical performance of biofunctionalised carbon-based nanomaterials are required.-The last aspect that has to be highlighted relates to biodegradability. Ideally, bone tissue-engineered scaffolds should be biodegradable; this opens up a new research avenue—to study the behaviour of carbon-based nanomaterials and the associated degradative by-products when implanted into the body. Some studies have appeared related to this issue, but were inconclusive and inconsistent results were found. Further research on cytotoxicity and possible adverse environmental effects is necessary before these scaffolds can be clinically tested.-Safety and success in clinical translation need to be demonstrated by facing the regulatory and economic hundles. However, the future of this technology is bright, and the commitment of scientists and engineers will lead to a fruitful and impactful future in the coming decades.

## Figures and Tables

**Figure 1 materials-13-05083-f001:**
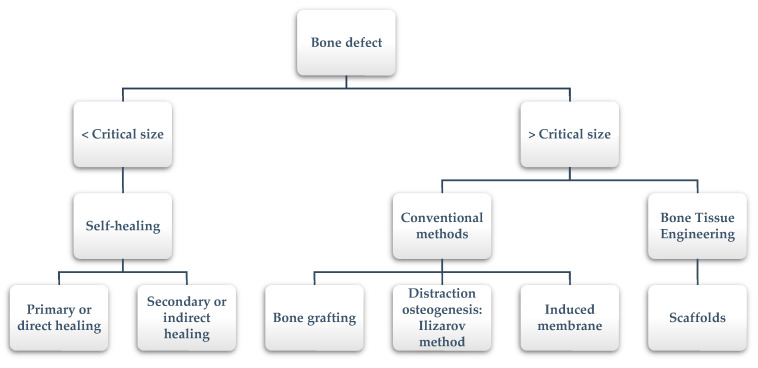
Bone defect healing: from self-healing to bone tissue engineering.

**Figure 2 materials-13-05083-f002:**
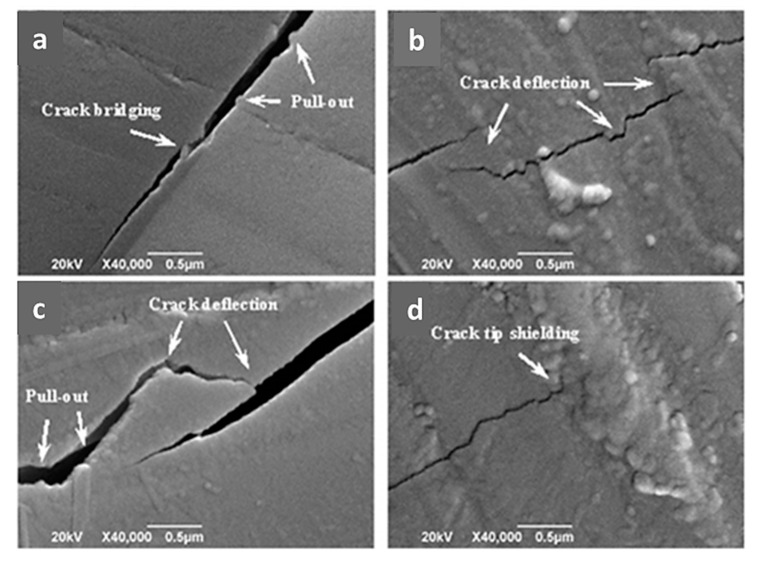
Inhibition of crack propagation by graphene in bioactive glass scaffold by different mechanisms: (**a**–**c**) crack deflection, crack bridging and graphene pull-out; (**d**) termination of crack growth at the crack tip [73].

**Figure 3 materials-13-05083-f003:**
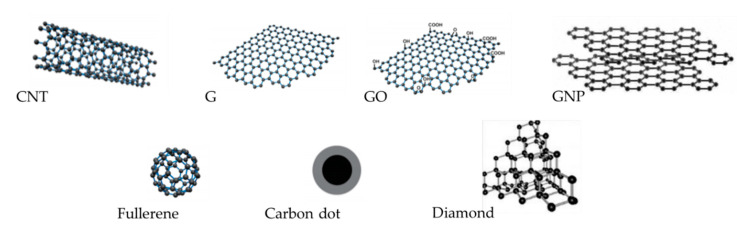
Carbon-based nanomaterials (modified from [117]).

**Figure 4 materials-13-05083-f004:**
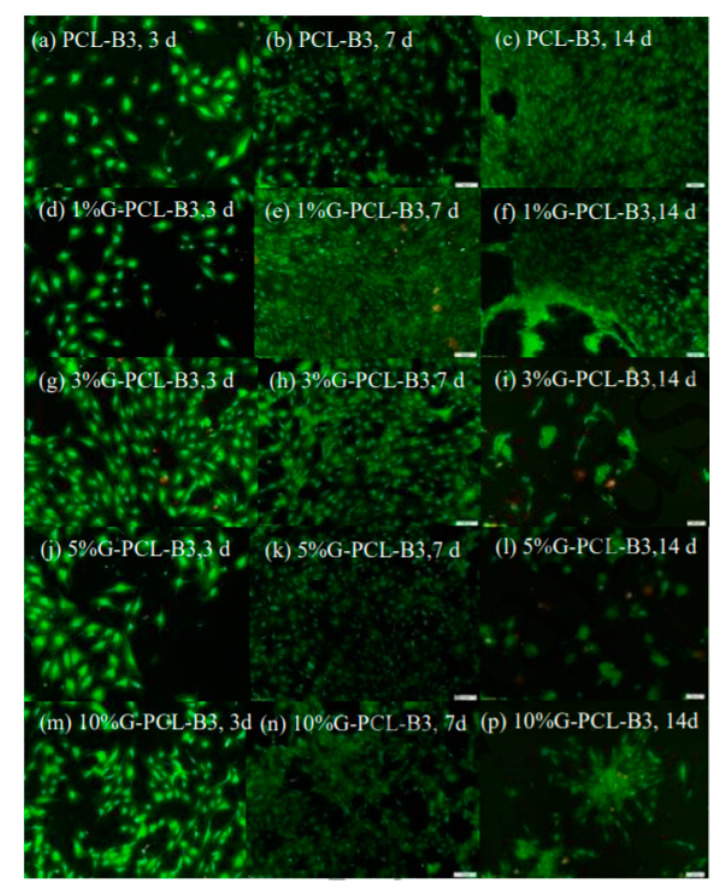
Live (green) and dead (red) MC3T3-E1 cells seeded on poly ε-caprolactone (PCL)-coated bioactive glass with different percentages of graphene. (**a**–**c**) Without graphene, (**d**–**f**) 1 wt.% graphene (G), (**g**–**i**) 3 wt.% G, (**j**–**l**) 5 wt.% G and (**m**–**p**) 10 wt.% G. After 7 days, the density of cells on G-containing scaffolds was higher than without G. However, after 14 days of incubation, a decrease in cell viability is observed due to the presence of G (reprinted from [170] with permission from Elsevier).

**Figure 5 materials-13-05083-f005:**
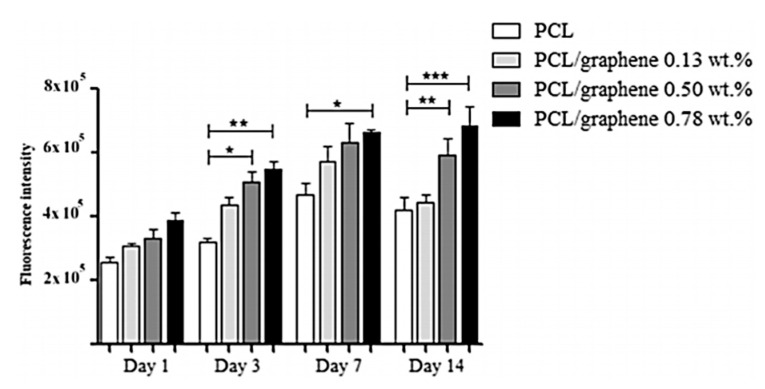
MC3T3 osteoblasts viability and proliferation measured by fluorescence intensity for G-containing PCL scaffolds. The higher the G content, the higher the cell proliferation rate. (*****) Statistical analysis difference *p* < 0.05; (******) *p* < 0.01; (*******) *p* < 0.001 (reprinted from [93], with permission from Elsevier).

**Figure 6 materials-13-05083-f006:**
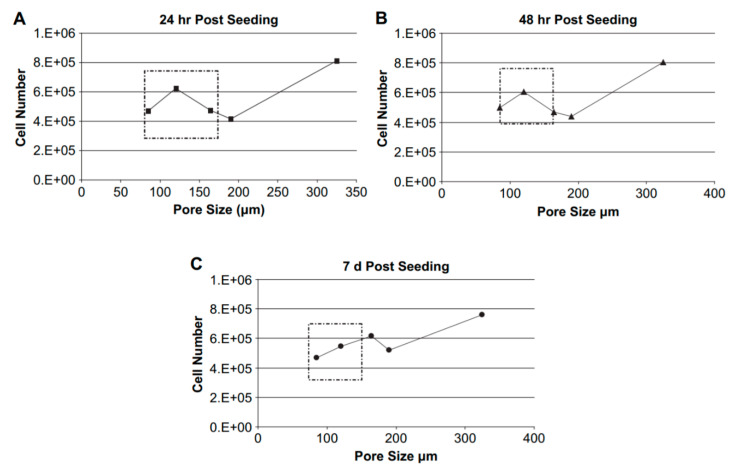
Effect of mean pore size of collagen-glycosaminoglycan scaffolds on MC3T3-E1 cell attachment and proliferation at different time points: at 24 h (**A**), at 48 h (**B**) and 7 days (**C**). The relation between mean pore size and cell response is non-linear (reprinted from [198], with permission from Elsevier).

**Figure 7 materials-13-05083-f007:**
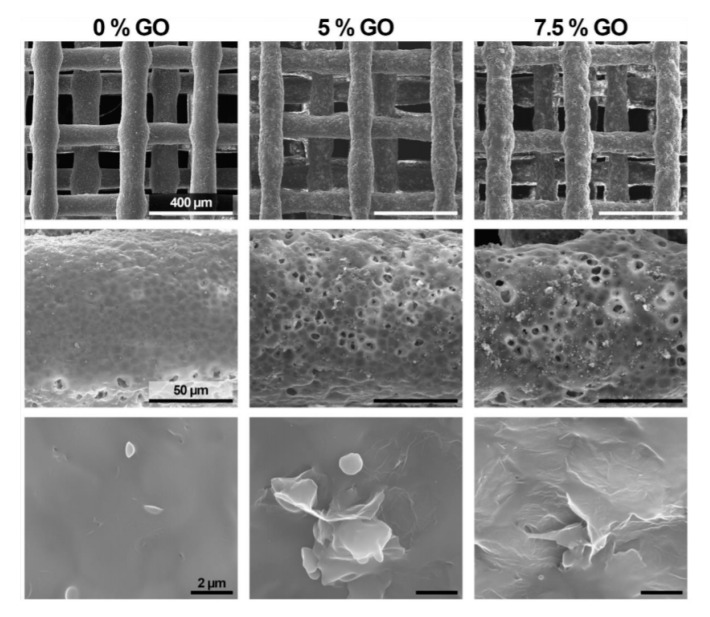
SEM images of PCL scaffolds with different percentages of Graphene oxide (GO), manufactured by Fused Deposition Modeling (FDM). Surface roughness and irregularity increased with the addition of GO. The average diameter of the fibres did not change with GO (reprinted from [260], with permission from Elsevier).

**Figure 8 materials-13-05083-f008:**
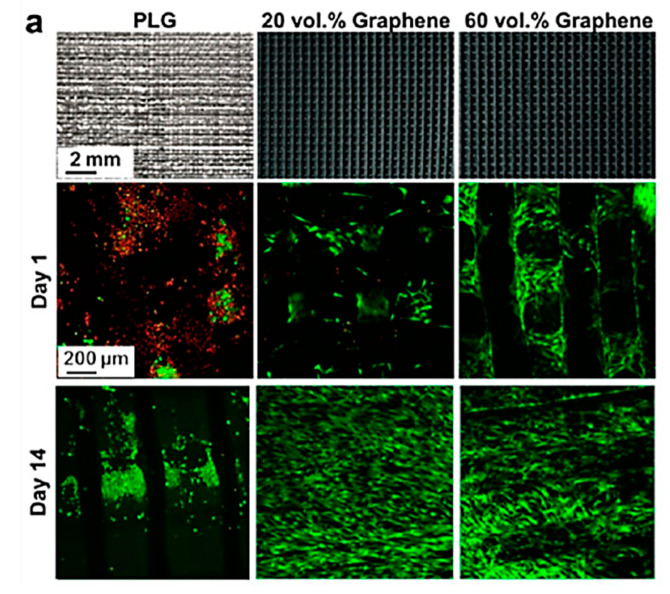
Photographs and cell viability (live cells in green and dead cells in red) of poly (lactic-co-glycolide) (PLG) scaffolds loaded with G manufactured by Direct Ink Writing (DIW). An increase in cell viability was produced when G was added (reprinted with permission from [268]. Copyright 2015 American Chemical Society).

**Figure 9 materials-13-05083-f009:**
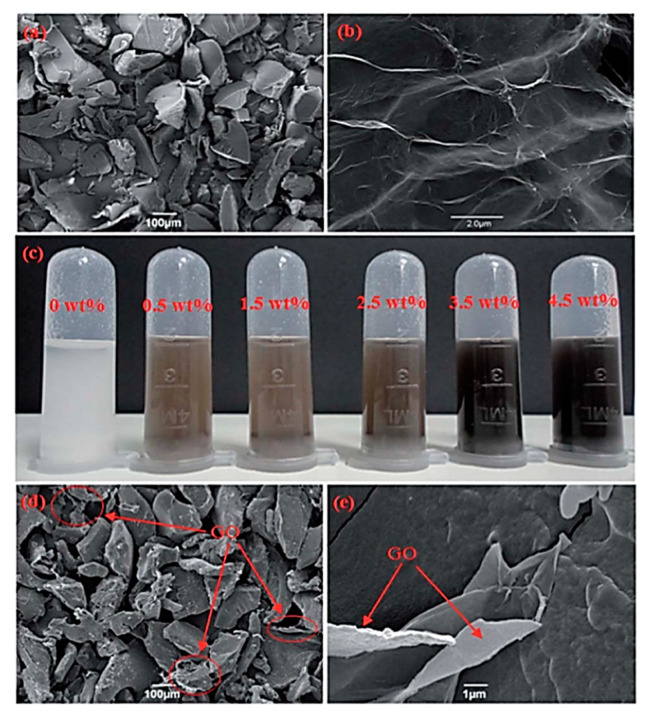
Preparation of poly(vinyl alcohol) (PVA)-GO powder for Selective Laser Sintering (SLS) printing: (**a**) SEM image of initial PVA powder; (**b**) TEM image of initial GO; (**c**) photographs of GO/PVA suspersion in deionized water after ultrasonication; (**d**,**e**) SEM images of the composite powder after evaporation of water [98].

**Figure 10 materials-13-05083-f010:**
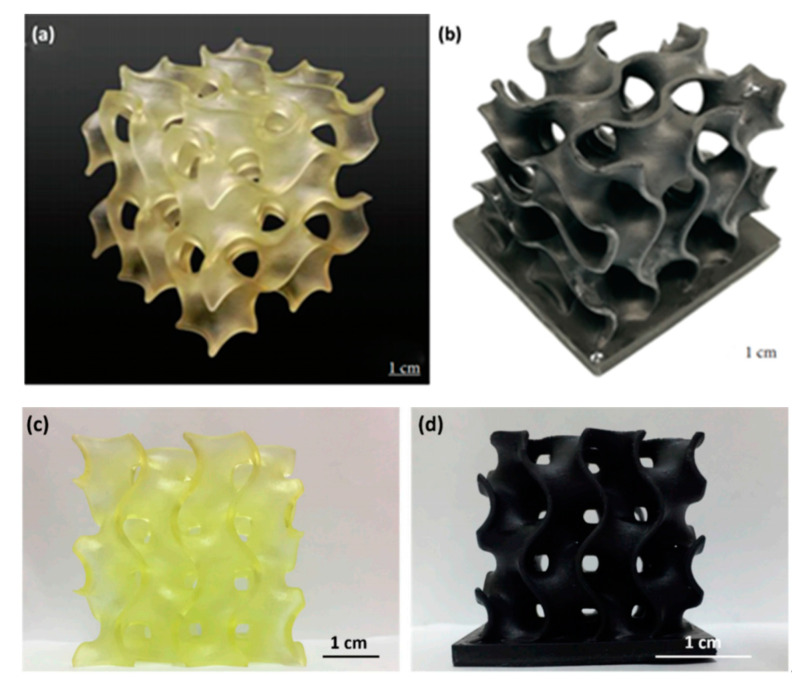
Polylactic acid (PLA)/PUA gyroid scaffolds manufactured by (**a**,**b**) Stereolithography (SLA) [238] and (**c**,**d**) Digital Light Processing (DLP) [240]. The addition of nanomaterials did not affect the printing process.

**Figure 11 materials-13-05083-f011:**
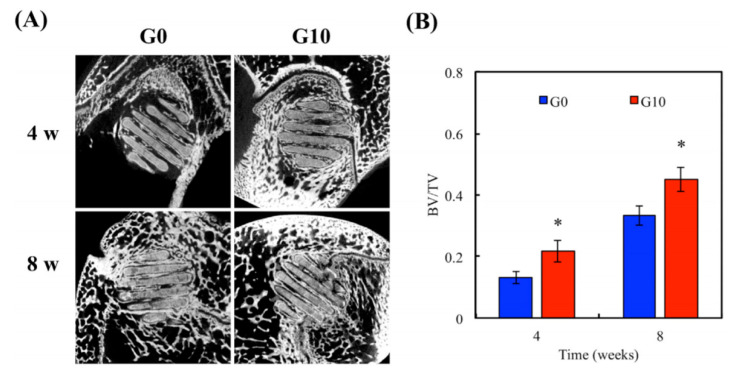
(**A**) Morphology of bone growth on CS/PCL scaffold with (G10) and without (G0) graphene manufactured by FDM. Images took by μ-CT. (**B**) Relative bone mass volume (BV/TV) at fixed-sized critical lesion ar different times. It is seen how the presence of G increased the bone growth rate (reprinted from [68], with permission from Elsevier).

**Figure 12 materials-13-05083-f012:**
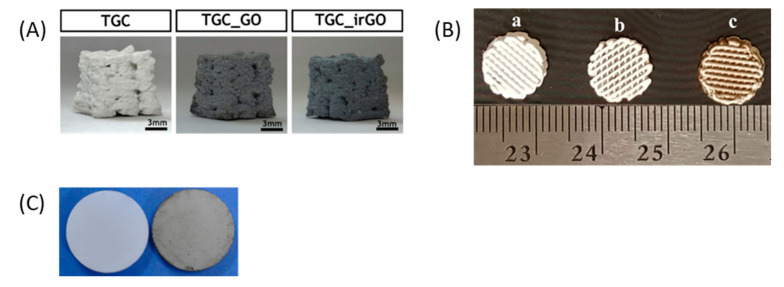
Ceramic parts printed by DIW with and without GO. (**A**) Images of scaffolds produced with GO added to the ink (reprinted from [70], with permission from Elsevier); (**B**) photographs of scaffolds produced (a) without nanomaterials, (b) with 0.25 wt.% and 0.75 wt.% of nanomaterials added by infiltration into the pores (reprinted from [69], with permission from Elsevier); (**C**) Tricalcium phosphate (TCP) disks without and with GO coating (reprinted from [54], with permission from Elsevier).

**Figure 13 materials-13-05083-f013:**
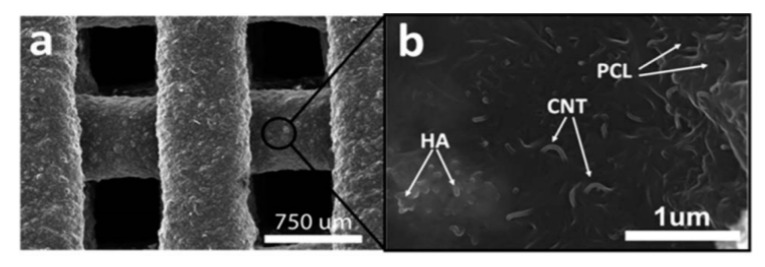
(**a**) SEM images of PCL/Hydroxyapatite (HA)/multiwalled carbon nanotube (MWCNT) scaffold fabricated using DIW. (**b**) Images with higher magnification. MWCNTs that were incorporated at a low level of loading were well-dispersed within the polymer matrix, which increased the mechanical properties (reprinted from [274], with permission from Elsevier).

**Figure 14 materials-13-05083-f014:**
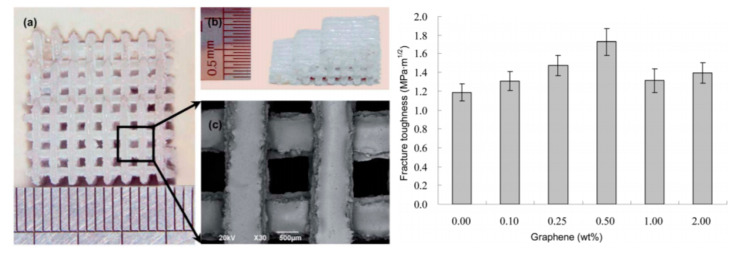
Images of Graphene-CaSiO_3_ porous scaffold fabricated using SLS (**left**) and variation in fracture toughness on addition of G (**right**). There was an optimal level of G loading in terms of fracture toughness due to the inability to disperse the G within the matrix [275].

**Table 1 materials-13-05083-t001:** Properties of carbon-based nanomaterials for bone tissue engineering applications.

	Bactericidal Mechanism	Osteoconductivity	Possible Toxicity	References
**Carbon nanotubes**	Binding	Electrochemical interactions with cells Increase wettability	Biopersistent Easy penetration in the cell membrane Metal catalysts trapped	[132,133,157,158,159,160,161,162,163,164,165]
**Graphene**	Physical damage	Increase wettability	Agglomeration on cell membranes ROS generation	[93,134,143,144,160,166,167,168,169,170]
**Graphene oxide**	Physical damage ROS generation	Increase wettability	ROS generation	[134,135,171,172,173,174,175,176]
**Fullerenes**	Binding	Only after aging	Induce DNA breakage ROS generation	[151,152,177,178,179,180]
**Nanodiamond**	Binding	Increase wettability	No cytotoxicity No ROS generation	[153,154,181,182,183,184]

**Table 2 materials-13-05083-t002:** Comparison of technologies of additive manufacturing available for biodegradable materials with nanomaterials.

	FDM	DIW	SLS	SLM	SLA	DLP
**Material**	Thermoplastic polymer	Polymer or polymer + ceramic	Polymer or ceramic	Thermoplastic polymer	Photocurable polymer	Photocurable polymer
**Morphology**	Filament	Ink	Powder	Powder	Liquid	Liquid
**Accuracy**	Low	Medium	Medium	Low	High	High
**Nanomaterial dispersion**	Solvent dissolution Melt mixing by extrusion	Solvent mixing Centrifuge mixing Ultrasonication	Melt mixing by extrusion Dissolution—precipitation Physical mixing	Melt mixing	Solvent mixing Ultrasonication	Ultrasonication
**Advantages**	Simplicity, speediness and large-scale production Low cost The most common AM technology	Flexible manufacturing Low cost Large parts manufacturing	High print speed Good mechanical properties Printing w/o support structures	Excellent mechanical properties Improved density compared to SLS Printing w/o support structures	Smooth surface finish Excellent part quality Excellent ability to fabricate complex structures UV sterilisation	High accuracy Excellent part quality Excellent ability to fabricate complex structures Higher print speed than SLA UV sterilisation
**Limitations**	Support structures required Highly anisotropic parts Nozzle clogging Layer delamination Sterilisation process may affect the material Pore size limited by the low accuracy	Support structures required Deposited ink should retain its shape	High temperature reached Unmelted powders may be trapped Extensive cleaning is needed after printing Polymer powder production with adequate flowability Expensive	High temperature reached Unmelted powders may be trapped Extensive cleaning is needed after printing Polymer powder production with adequate flowability Pore size limited by the low accuracy	Extensive post-treatments Uncured resin toxicity Support structures required Resin cannot be storage indefinitely	Extensive post-treatments Uncured resin toxicity Support structures required Resin cannot be storage indefinitely
**Ref**	[218,225,226,227,228]	[218,229,230]	[218,231,232,233]	[218,234,235]	[218,236,237,238,239]	[218,240]

**Table 3 materials-13-05083-t003:** Polymer scaffolds obtained by 3D printer with different carbon-based nanomaterials.

Technology	Nanomaterial	Nanomaterial Dispersion	Matrix	Effect of Carbon-Based Nanomaterials	Ref.
FDM	GO (0.5 wt.%)	Solvent mixing	TPU/PLA	Increase tensile and compression modulus Low amount of GO increases cell proliferation	[65]
	GO (0.3 wt.%)	Solvent mixing	PLA	Increase Young’s modulus Increase toughness More efficient promotion of cell adhesion and proliferation	[262]
	GO (7.5 wt.%/ 0.5 wt.%)	Solvent mixing	PCL	Improve antimicrobial properties Enhancement of cellular response	[260,261]
	G (0.78 wt.%)	Melt mixing	PCL	Cell proliferation stimulation Increase hydrophilicity Increase compressive modulus and strength	[93,143,263]
	G (3 wt.%)	Solvent mixing	PTMC	Increase electrical conductivity Increase tensile strength, elongation at break and Young’s modulus No effect on cell attachment and viability	[264]
	GNP (4 wt.%)	Melt mixing	ABS	Increase tensile modulus Reduction in ultimate tensile stress and strain Reduction in creep compliance	[227]
	Carbon nanofibers/GNP (18 wt.%)	Commercial filament	PLA	CNF reduces compression stiffness Improve bioactivity	[265]
	MWCNT (3 wt.%)	Melt mixing	PCL	Increase in compressive modulus and strength Improve cell viability and proliferation Increase polymer crystallinity Increase hardness and elastic modulus	[263,266]
	MWCNT/nHA (0.75 wt.%)	Melt mixing	PCL	Increase compressive strength Improve cell attachment	[267]
DIW	G (20 vol.%)	Solvent mixing	PLG	Increase tensile modulus High loading decreases the tensile strength Increase cell proliferation	[268]
SLS	GO (2.5 wt.%)	Ultrasonication of water dispersion	PVA	Increase tensile strength, elongation at break, compressive modulus and compressive strength Good cytocompatibility	[98]
	GO (1 wt.%)	Ultrasonication of water dispersion	PEEK */ PVA	Increase surface energy Increase compressive modulus and strength Increase cell proliferation	[269]
	GO (1 wt.%)	Ultrasonication of solvent dispersion	PLLA	Increase compressive strength Increase hardness	[270]
	GO/Ag (1 wt.%)	Ultrasonication of solvent dispersion + ball milling	PLLA/ PGA	Increase compressive strength and modulus Increase wettability Antibacterial effect	[271]
	GO (0.3 wt.%)	Ultrasonication of solvent dispersion	PVDF *	Increase compressive strength, tensile strength, and modulus Increase hydrophilicity Improve cellular response	[99]
	ND/MoS_2_ (2 wt.%)	Ultrasonication of solvent dispersion	PHBV	Increase tensile strength and modulus Increase compressive strength and modulus Enhanced mineral deposition	[272]
SLA	G (0.5 wt.%)	Ultrasonication	PLA/ PUA	Increase tensile strength Increase flexural strength and modulus	[238]
DLP	GNP (0.5 wt.%)	Ultrasonication	PLA/ PUA	Increase flexural modulus and fracture toughness No effect on printability	[240]

* Non-biodegradable.

**Table 4 materials-13-05083-t004:** Ceramic-based 3D scaffolds fabricated by using additive manufacturing (AM) techniques with different carbon-based nanomaterials.

Technology	Nanomaterial	Nanomaterial Dispersion	Matrix	Effect of Carbon-Based Nanomaterial	Ref.
FDM	G (10 wt.%)	Ultrasonication of solvent dispersion	Calcium silicate/PCL	Increase hydrophilicity Increase Young’s modulus Increase compressive strength Improve cellular response and bone regeneration	[68]
DIW	GO (50 wt./vol.%)	Coating prepared with water/GO suspension ultrasonically stirred	β-TCP/PVA	Enhanced biological properties: cell proliferation, alkaline phosphatase activity and osteogenic gene expression	[69]
	GO (0.5 wt.%)	Mechanical stirring	TCP/AA	Improve compressive and biological performance	[174]
	GO/rGO (0.3 wt.%)	Mechanical stirring	TCP/gelatin/chitosan	Both rGO and GO increase compressive strength and modulus. rGO has more effect. GO improves calcium deposition	[70]
	rGO/Mg (50% wt./vol)	Filled into the pores of 3D printed scaffolds	β-TCP/carboxymethylcellulose/sodium tripolyphosphate	Increase surface roughness Increase Young’s modulus Improve cell proliferation Lower doses increase osteogenic differentiation	[54]
	G (21–24.5 vol.%)	Mixed by hand with solvent	HA/PLGA	Compared with HA, graphene reduces compressive modulus and increases strain to failure	[273]
	MWCNT (0.75 wt.%)	Solvent mixing	HA/PCL	Increase compressive strength for low content of CNT Improve cell attachment Increase electrical conductivity Reduce compressive modulus	[274]
SLS	G (0.5 wt.%)	Ultrasonication of solvent dispersion	Nano-58S bioactive glass	Improve compressive strength Improve fracture toughness	[73]
	G (0.5 wt.%)	Ultrasonication of solvent dispersion + ball milling	Calcium silicate	Improve compressive strength Improve fracture toughness	[275]
	GNP (1 wt.%)	Ultrasonication of solvent dispersion	Di	Improve compressive strength Improve fracture toughness	[276]
	MWCNT (3 wt.%)	Ultrasonication of solvent dispersion + ball milling	13–93 bioactive glass	Improve compressive strength Improve fracture toughness	[277]
	MWCNT (2 wt.%)	Ultrasonication of solvent dispersion	Di	Improve compressive strength Improve fracture toughness	[278]
	CNT (0.2 wt.%)	Mechanical mixing	β-TCP	Improve compressive strength	[279]
	CNT/GNP (2 wt.%)	Ultrasonication of solvent dispersion	Di	Synergistic effect of nanomaterials Increase compressive strength and modulus Good cytocompatibility	[280]

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
