# Peer review of "Advances in Biodegradable 3D Printed Scaffolds with Carbon-Based Nanomaterials for Bone Regeneration"

_materials, 2020, doi:10.3390/ma13225083_

Round 1
Reviewer 1 Report
The authors were aimed to explore the potential research opportunities and challenges of 3D printed biodegradable composite-based scaffolds containing carbon-based nanomaterials for bone tissue engineering applications. The study covers some issues that have been overlooked in other similar topics. The structure of the review manuscript appears adequate and well divided in the sub-paragraphs.
The paper deserves publication after addressing the following changes:
- Please add a sentence about the interplay of mesenchymal stem cells and bioengineered scaffolds (see an discuss: PMID: 29692270 and PMID: 29445404)
- The conclusions is too long and should give useful outcomes to the readers and not only summarize the main steps of the study.
Author Response
Dear Reviewer:
We are sending you the revised version of the manuscript entitled:
"Advances in biodegradable 3D printed scaffolds with carbon-based nanomaterials for bone regeneration”, Manuscript ID: materials-983934 prepared by S. Lopez de Armentia, J.C. del Real, E. Paz and N.J. Dunne.
We thank you for the careful reading of the manuscript and your valuable comments that have helped us to improve it.
We hope to have answered all the queries raised and that the manuscript is now acceptable for publication in Materials journal. Your comments followed by the corresponding answers are shown as follows:
The authors were aimed to explore the potential research opportunities and challenges of 3D printed biodegradable composite-based scaffolds containing carbon-based nanomaterials for bone tissue engineering applications. The study covers some issues that have been overlooked in other similar topics. The structure of the review manuscript appears adequate and well divided in the sub-paragraphs.
The paper deserves publication after addressing the following changes:
- Please add a sentence about the interplay of mesenchymal stem cells and bioengineered scaffolds (see an discuss: PMID: 29692270 and PMID: 29445404)
An explanation and the references have been added to the manuscript (Lines 518-520).
- The conclusions is too long and should give useful outcomes to the readers and not only summarize the main steps of the study.
Conclusions has been reduced and some outcomes has been added.
Reviewer 2 Report
In this review, Lopez de Armentia et al. summarized advances in biodegradable 3D printed scaffolds and discussed its application for bone tissue engineering applications. This review is comprehensive and well written. Also, educational not only for this research field specialists but also for a broad range of readership. I do not have any criticisms for this review. I believe this review will contribute to this research field. I have just one minor comment to this review.
In Figure 2, it is better to describe what each image are in the legend although the authors cited original paper.
Author Response
Dear Reviewer:
We are sending you the revised version of the manuscript entitled:
"Advances in biodegradable 3D printed scaffolds with carbon-based nanomaterials for bone regeneration”, Manuscript ID: materials-983934 prepared by S. Lopez de Armentia, J.C. del Real, E. Paz and N.J. Dunne.
We thank you for the careful reading of the manuscript and your valuable comments that have helped us to improve it.
We hope to have answered all the queries raised and that the manuscript is now acceptable for publication in Materials journal. Your comments followed by the corresponding answers are shown as follows:
In this review, Lopez de Armentia et al. summarized advances in biodegradable 3D printed scaffolds and discussed its application for bone tissue engineering applications. This review is comprehensive and well written. Also, educational not only for this research field specialists but also for a broad range of readership. I do not have any criticisms for this review. I believe this review will contribute to this research field. I have just one minor comment to this review.
In Figure 2, it is better to describe what each image are in the legend although the authors cited original paper.
Figure 2 legend has been changed and an explanation of each imagen has been added.
Reviewer 3 Report
Dearest Authors,
very interesting review on a becoming hot topic in this field.
IMHO I would like to suggest two minor improvements:
1) it would be useful to underline the potential translability into clinics of any of these futurable technologies, eventually also outlining regulatory pathway etc (at least in major systems such as US-FDA and EU-MDR);
2) the new international paradigm of research & development in the field of "nano" polymeric systems is the so called "safety by design". it should be precisely quoted and referred to. On chitosan NPs, for examples, the works by Prof. G. Borchard (Uni Geneve) and Dr. Peter Wick (Empa) should be quoted and referenced.
More detailed comments:
1) literature suggets very often new possible bomne grafts and bone substitutes biomaterials, but only 1 out 100 really sees the clinics. It hence would be very useful to underline the potential translability into clinics of those new materials described. Eventually the authors could comment on this and provide theiropinion on these futurable technologies. They could also eventually critically comment if and which of these technology can be used according to the major regulatory pathways of US-FDA and EU-MDR.
2) "nano" biomaterials often poses safety related issues. The new international paradigm of research & development in the field of "nano" polymeric systems is the so called "safety by design". it should be precisely quoted and referred to. On chitosan NPs, for examples, the works by Prof. G. Borchard (Uni Geneve) and Dr. Peter Wick (Empa) should be quoted and referenced.
best & stay safe
Author Response
Dear Reviewer:
We are sending you the revised version of the manuscript entitled:
"Advances in biodegradable 3D printed scaffolds with carbon-based nanomaterials for bone regeneration”, Manuscript ID: materials-983934 prepared by S. Lopez de Armentia, J.C. del Real, E. Paz and N.J. Dunne.
We thank you for the careful reading of the manuscript and your valuable comments that have helped us to improve it.
We hope to have answered all the queries raised and that the manuscript is now acceptable for publication in Materials journal. Your comments followed by the corresponding answers are shown as follows:
Dearest Authors,
very interesting review on a becoming hot topic in this field.
IMHO I would like to suggest two minor improvements:
1) it would be useful to underline the potential translability into clinics of any of these futurable technologies, eventually also outlining regulatory pathway etc (at least in major systems such as US-FDA and EU-MDR);
We found this comment very interesting, so a section with this topic has been added (lines 924-952).
2) the new international paradigm of research & development in the field of "nano" polymeric systems is the so called "safety by design". it should be precisely quoted and referred to. On chitosan NPs, for examples, the works by Prof. G. Borchard (Uni Geneve) and Dr. Peter Wick (Empa) should be quoted and referenced.
A paragraph regarding safety by design has been added in lines 481-488.